# A gRNA-tRNA array for CRISPR-Cas9 based rapid multiplexed genome editing in *Saccharomyces cerevisiae*

Yueping Zhang [1], Juan Wang[1], Zibai Wang[1], Yiming Zhang[1], Shuobo Shi[1], Jens Nielsen [1,2,3] & Zihe Liu[1]

With rapid progress in DNA synthesis and sequencing, strain engineering starts to be the rate-limiting step in synthetic biology. Here, we report a gRNA-tRNA array for CRISPR-Cas9 (GTR-CRISPR) for multiplexed engineering of *Saccharomyces cerevisiae*. Using reported gRNAs shown to be effective, this system enables simultaneous disruption of 8 genes with 87% efficiency. We further report an accelerated Lightning GTR-CRISPR that avoids the cloning step in *Escherichia coli* by directly transforming the Golden Gate reaction mix to yeast. This approach enables disruption of 6 genes in 3 days with 60% efficiency using reported gRNAs and 23% using un-optimized gRNAs. Moreover, we applied the Lightning GTR-CRISPR to simplify yeast lipid networks, resulting in a 30-fold increase in free fatty acid production in 10 days using just two-round deletions of eight previously identified genes. The GTR-CRISPR should be an invaluable addition to the toolbox of synthetic biology and automation.

[1] Beijing Advanced Innovation Center for Soft Matter Science and Engineering, College of Life Science and Technology, State Key Laboratory of Chemical Resource Engineering, Beijing University of Chemical Technology, 100029 Beijing, China. [2] Department of Biology and Biological Engineering, Chalmers University of Technology, SE412 96 Gothenburg, Sweden. [3] Novo Nordisk Foundation Center for Biosustainability, Technical University of Denmark, DK2800 Lyngby, Denmark. These authors contributed equally: Yueping Zhang, Juan Wang. Correspondence and requests for materials should be addressed to Z.L. (email: zihe@mail.buct.edu.cn)

Genome editing, which requires precise DNA changes at predetermined locations, plays a crucial role in metabolic engineering and synthetic biology. Given our limited knowledge of complex cellular networks, intensive engineering at multiple genomic loci is often required in both basic and applied research. The recently developed Clustered Regularly Interspaced Short Palindromic Repeats (CRISPR) system has greatly accelerated the speed of strain engineering in a wide range of organisms[1–4]. However, its multiplexed application is still limited by its guide RNA (gRNA) processing efficiency and throughput[1,5].

Generally, two approaches have been applied for expressing multiple gRNAs: one approach is to transcribe each gRNA cassette with individual RNA polymerase promoter, and the other is to use one single promoter to transcribe all gRNAs in one single transcript, which is then processed through different strategies to release individual gRNAs. These strategies require that each gRNA is flanked with cleavable RNA sequences, such as self-cleavable ribozyme sequences (e.g. Hammerhead ribozyme and HDV ribozyme)[6,7], exogenous cleavage factor recognition sequences (e.g. Cys4)[8], and endogenous RNA processing sequences (e.g. tRNA sequences and introns)[9–13]. Currently, in *Saccharomyces cerevisiae*, the expression of a maximum five gRNAs on one construct has been reported[5]. Developing strategies to further increase the efficiencies of multiplexed gene editing would benefit both applied research such as constructing metabolic cell factories and basic research such as generating minimal genomes.

In addition, most present multiplexed CRISPR-Cas9 systems lack fast and facile procedures and require plasmid cloning steps in *Escherichia coli*, as shown in Table 1. Moreover, these systems often need preintegration of the Cas9 gene into the chromosome, construction of helper plasmids for gRNA assembly, or gene synthesis of multiple gRNAs[5,8,12,14], which are laborious and time-consuming. Therefore, multiplexed genome-editing techniques that eliminate the cloning step in *E. coli* are advantageous, especially for applications in automated multiplexed genome editing and combinatorial transcriptional regulations. So far, similar concepts have been reported for targeting 1–3 genome loci (Table 1); however, such methods often suffer from low efficiencies, limited gRNA numbers, and problem of non-equimolar gRNA expression[1,15–17]. Indeed, to develop such systems are challenging with several key limitations, for example, the low plasmid construction efficiency with compact repetitive sequences (the gRNA scaffold sequences in this case), the very high

homology recombination efficiency of *S. cerevisiae* that could uncontrollably circularize the unassembled fragments with repetitive sequences, and the low throughput procedures of current multiplexed systems.

Here we present a gRNA-tRNA array for CRISPR-Cas9 (GTR-CRISPR) that allows simultaneous disruptions of 8 genes in *S. cerevisiae* with 87% efficiency. We further report a Lightning GTR-CRISPR, which skips the *E. coli* transformation and verification steps by directly transforming the Golden Gate reaction mix to yeast. Using reported gRNAs shown to be effective, we achieved 4-gene disruptions at 96% efficiency and 6-gene disruptions at 60% efficiency in 3 days. For un-optimized gRNAs, the Lightning GTR-CRISPR enabled 6-*HIS*-gene disruptions at 23% efficiency in 3 days. This system greatly accelerates the speed of yeast strain development, and we achieved an increase in free fatty acid (FFA) production by about 30-fold in 10 days using just 2-round deletions of 8 previously identified genes.

## Results

**Evaluations of gRNA expression systems**. We chose the *S. cerevisiae* endogenous tRNA$^{Gly}$ for gRNA processing (Fig. 1) and used it to compare different gRNA expression systems. The reasons for choosing tRNA$^{Gly}$ are (i) tRNA$^{Gly}$ has been applied in gRNA processing in plants and *Drosophila*[9–11] and (ii) tRNA-$^{Gly}$ gene is relatively short with 71 base pairs (bp) compared to other endogenous tRNAs, and this enables a simple and compact architecture, so that gRNAs may be transcribed more efficiently.

We first tested three different gRNA expression modes for introducing simultaneous disruptions of 3–5 genes in *S. cerevisiae* (Fig. 2a). Mode A was a single transcript of gRNA-tRNA$^{Gly}$ array (GTR) under a widely used RNA polymerase III promoter, *SNR52* promoter[5,18]; mode B contained multiple gRNA expression cassettes with each cassette composed of one *SNR52* promoter, one gRNA, and one *SNR52* terminator; and mode C also contained multiple gRNA expression cassettes, and from the second cassette on, each cassette was composed of one fusion promoter of *SNR52* promoter and tRNA$^{Gly}$ sequence, one gRNA, and one *SNR52* terminator. In order to avoid the gRNA efficiency bias and to get a quick comparison of our system with the reported multiplexed CRISPR-Cas9 systems in *S. cerevisiae*, the gene disruption targets and gRNA sequences used were selected from published papers except for stated otherwise[8,14]. The 100 bp DNA homologous recombination templates (donors) were

## Table 1 Comparison of different CRISPR-Cas9-based multiplexed gene-editing systems

| CRISPR systems | Efficiency | gRNA processing | Cas9 pre-transformation | Required additional procedures | Total time spent[a] |
|---|---|---|---|---|---|
| **Plasmid-required systems** | | | | | |
| HI-CRISPR[14] | 3 targets with 100% | Endogenous | No | Gene synthesis | 10–13 days |
| CRISPRm[7] | 1–3 targets with 81–100% | Self-cleavable ribozymes | No | Helper plasmid | 8–10 days |
| Csy4-based CRISPR[8] | 4 targets with 96% | Csy4 cleavage | Yes | Helper plasmid | 11–13 days |
| CasEMBLR[5,26] | 1–5 targets with 50–100% | Individual cassette | Yes | Helper plasmid | 11–13 days |
| CRISPR by Mans and Rossum et al.[15] | 6 targets with 65% | Individual cassette | Yes | 3 plasmids for 6 gRNAs | 8–9 days |
| GTR-CRISPR (this work) | 8 targets with 87% | tRNA processing | No | No | 6–7 days |
| **Cloning-free systems** | | | | | |
| CRISPR by Generoso et al.[16] | 2 targets with <15% | Individual cassette | No | No | 2 days |
| CAM[17,27] | 3 targets with 64% | Individual cassette | Yes | No | 5 days |
| Lightning GTR-CRISPR (this work) | 4 targets with 96% 6 targets with 60% | tRNA processing | No | No | 3 days |

*CRISPR* Clustered Regularly Interspaced Short Palindromic Repeats, *gRNA* guide RNA
[a]Total time was calculated including Cas9 pre-transformation, gene synthesis, plasmid construction, and yeast transformation (details as shown in Supplementary Table 2)

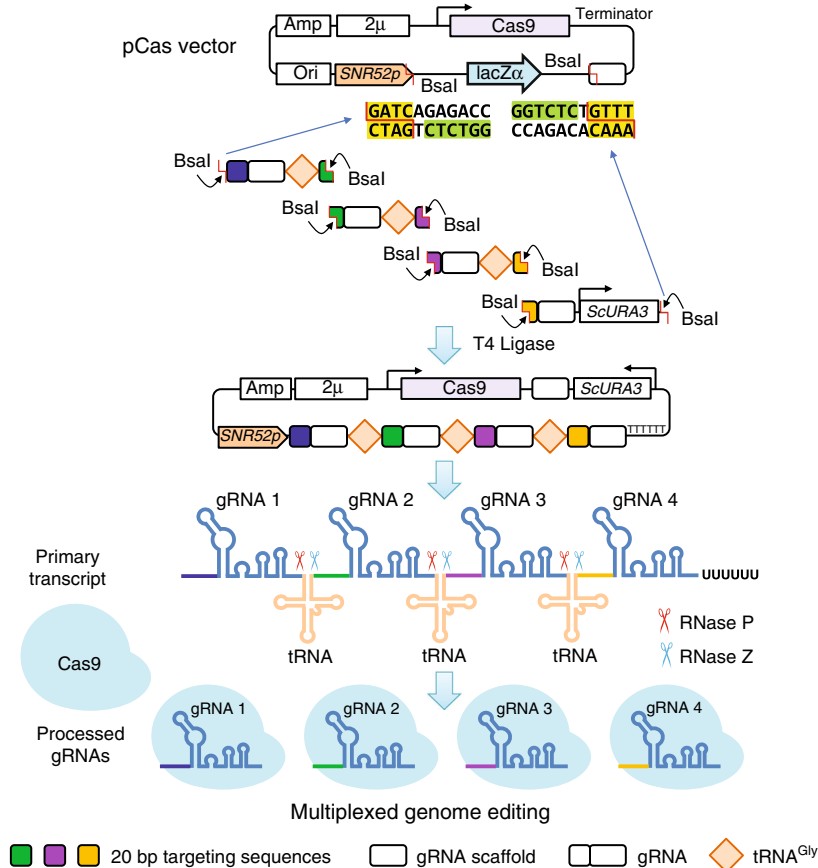

**Fig. 1** The GTR-CRISPR for multiplexed genome editing in *S. cerevisiae*. The GTR-CRISPR plasmid is constructed by Golden Gate assembly of PCR-generated fragments containing Cas9, gRNA-tRNA-array cassette, and a *S. cerevisiae URA3* gene with the truncated promoter. After transformed to yeast, the primary RNA transcript is cleaved by yeast endogenous tRNA-processing enzymes, RNase P and Z, leaving functional guide RNAs for genome editing

generated by PCR amplification of two ~60 bp oligonucleotide primers and co-transformed with the CRISPR plasmid. The donors will introduce 8-bp deletions around PAM sequences and generate frame-shifting gene disruptions. We chose to delete 8 bp including the PAM sequencing to avoid potential off-targets cleavage, which still may occur with 3–5 bp mismatches in the PAM-distal part of the gRNA sequence[14,19,20].

As shown in Fig. 2, for 3-gene disruptions, the overall efficiencies of modes A, B, and C were comparable with each other, i.e. 93.0%, 76.9%, and 80.3%, respectively, whereas for 4-gene and 5-gene disruptions, the overall efficiencies of mode A is significantly better than mode B and C. The disruption efficiencies for 4 genes with modes A, B, and C reached 100%, 18.2%, and 75.9%, respectively; and for 5-gene disruptions, the efficiencies of modes A, B, and C reached 88.9%, 6.7%, and 33.3%, respectively. The disruption efficiencies of mode C were significantly higher than mode B, suggesting that the fusion of tRNA$^{Gly}$ to the *SNR52* promoter indeed improved gene disruption efficiencies. This may be due to the consensus elements of the tRNA gene working as transcriptional enhancers for the transcription of RNA polymerase III[17,19].

**Characterization of the GTR-CRISPR system**. We demonstrated above that the GTR-CRISPR (mode A) can mediate efficient simultaneous disruptions of five genes. To further characterize the upper limits of gRNAs in single transcript, GTR-CRISPR plasmids containing 6–8 gRNAs in one transcript were constructed and transformed to yeast (Fig. 3a). The simultaneous disruption efficiencies for 6, 7, and 8 genes using the single GTR

transcript were 63.3%, 70%, and 36.5% respectively. These results showed that, when the targeting number is above five on one GTR transcript, the disruption efficiencies decreased sharply and the clonal variation increased significantly. This may be due to the insufficient transcription efficiency of the *SNR52* promoter when the transcript gRNA number increased to more than five.

To improve the targeting efficiency, we enhanced the strength of the promoter by fusing the *SNR52* promoter with the tRNA$^{Gly}$ sequence. However, when testing the disruption efficiency of the fusion promoter for expressing six gRNAs in GTR-CRISPR, the result showed no significant difference compared with the unfused *SNR52* promoter (Fig. 3b, A'6 vs. A6). As another strategy to improve the targeting efficiency, we introduced a second promoter as we have shown that 1 *SNR52* promoter can efficiently transcribe one GTR transcript containing 3–5 gRNAs. Indeed, the 2-promoter GTR-CRISPR system enhanced the disruption efficiency from 63.3% to 80% for 6 targets (Fig. 3b, A3A'3). Thus we further tested the 2-promoter GTR-CRISPR for disruptions of 8 genes, and the disruption efficiency was significantly improved from 36.5% to 86.7% (Fig. 3c). This is by far the largest number of efficient multiplexed disruptions for *S. cerevisiae*.

**The Lightning GTR-CRISPR without *E. coli* transformation**. We demonstrated that the GTR-CRISPR is able to disrupt up to eight genes with high efficiencies. This system does not need to perform gene synthesis, sub-cloning, pre-Cas9 integration, or transforming a separated Cas9 plasmid that already saves time and efforts comparing to most reported multiplexed CRISPR systems in yeast (Table 1). However, as most yeast

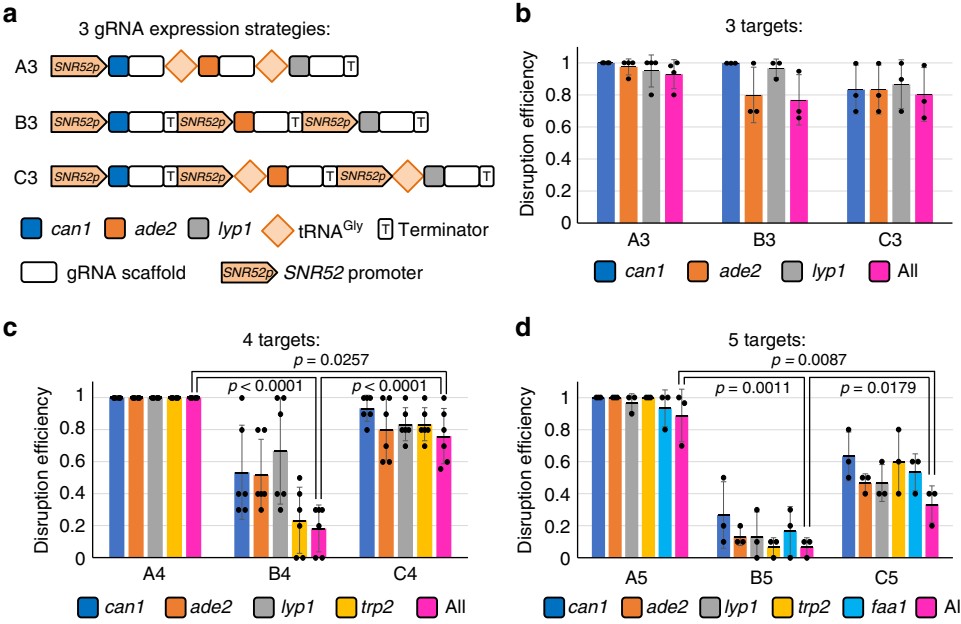

**Fig. 2** Evaluations of different guide RNA (gRNA) expression systems for 3–5 genes. **a** Graphic representation of the three different gRNA expression systems for three-gene disruptions. The A3, B3, and C3 represent three gRNAs expressed from mode A, B, and C, respectively. **b**–**d** Results of disruption efficiencies of 3–5 genes. The data of bar charts represent mean averages of each gene or overall disruption efficiencies. Each black dot represents the gene disruption efficiency of 10 colonies from each biological replicate, and the error bars indicate the standard deviations of all ($n \geq 3$) biological replicates. The statistical analyses were performed using unpaired $t$ Test. Source data are provided as a Source Data file

genome-editing systems, the GTR-CRISPR still requires to first transform the assembled plasmid to *E. coli* for plasmid amplification and verification, which takes about 4 days for PCR amplification, Golden Gate assembly, *E. coli* transformation, plasmid extraction, and DNA sequencing. To further reduce the time and simplify the procedure, we tested whether we could eliminate the *E. coli* cloning step by directly transforming the Golden Gate reaction mix to yeast (named the Lightning GTR-CRISPR; Fig. 4a).

We initially tested such procedure for four gRNAs in one GTR transcript (A4 in Fig. 4a), because of the high fidelity of plasmid construction in *E. coli* (10/10 *E. coli* colonies tested) and high yeast disruption efficiency (100%). However, after transforming the Golden Gate reaction mix with donors to yeast, we got a surprisingly low efficiency of simultaneous 4-gene disruptions at 7.1% (Fig. 4c, A4-ScU). This could be due to two reasons: (i) The endogenous *ura3-52* gene of yeast strain CEN. PK 113-5D might be repaired with the unassembled *URA3* marker fragment (*ScURA3*) in the Golden Gate reaction mix. Indeed, when we transformed the *ScURA3* fragment to yeast, a large number of cells grew on the selection plate. To overcome this problem, we changed the selection marker from the *ScURA3* to a heterologous *URA3* from *Kluyveromyces lactis* (*KlURA3*) with its endogenous 300 bp promoter (A4-KlU300 in Fig. 4b)[21] and got an increase of the 4-gene disruption efficiency from 7.1% to 16.4% (A4-KlU300 in Fig. 4c). (ii) The other reason may be resulting from homologous recombination among different gRNA fragments to be assembled, because of the existence of a high number of repetitive sequences (for each target to be disrupted, one gRNA scaffold and one tRNA$^{Gly}$ are required). Unlike verified plasmids, the Golden Gate reaction mix also has a large number of unassembled fragments, and transforming these fragments may cause yeast homologous repair, which results in truncated versions of the plasmid. After carefully examining the disruption efficiencies of these 4 genes, we found that the first gRNA targeting *CAN1* and the last gRNA targeting *TRP2* had much higher disruption efficiencies (97.5% and 37.5%, respectively)

than the efficiencies of the 2 targets in the middle (12.5% and 22.5%), whereas the disruption efficiencies of different gRNAs were similar in the GTR-CRISPR system with constructed plasmid (A4-ScU in Fig. 4c vs. A4 in Fig. 2c). We solved this problem by adopting an alternative assembly format with two *SNR52* promoters and each promoter expressing two gRNA (GTR format) with the *KlURA3* marker in the middle (A2A2-KlU300 in Fig. 4b and detailed in Supplementary Figure 1), so that the undesired homologous recombination would result in no marker on the plasmid and thus would not grow on the selection plate. Indeed, after transforming this Golden Gate reaction mix with donors to yeast, the efficiency of 4-gene disruptions was significantly improved from 16.4% to 71.7%. To further improve the disruption efficiency, we sought out to increase the copy number of the 2μ plasmid by truncating the promoter length of *KlURA3* from 300 bp to 200 bp and 100 bp. The efficiency of *KlURA3* with its 200 bp promoter was not significantly changed, whereas the efficiency of *KlURA3* with its 100 bp promoter was greatly improved to 95.6% for simultaneous disruptions of 4 genes (Fig. 4c, A2A2-KlU100). Real-time quantitative PCR analysis of these promoters was carried out and indicated that, indeed among these three promoters, the 100 bp promoter drives the least expression (Supplementary Figure 2b).

To further characterize the Lightning GTR-CRISPR system, we tested whether this system could simultaneously disrupt six genes. To avoid the homologous recombination among the different gRNA fragments, a second selection marker *LEU2* from *S. cerevisiae* (*ScLEU2*) and a third *SNR52* promoter were applied for two more gRNAs (Fig. 4b, A2A2A2), and yeast cells with additional *LEU2* gene deleted were used for the transformation. As shown in Fig. 4d, the Lightning GTR-CRISPR system was capable of simultaneous 6-gene disruptions with a 60% efficiency in 3 days. We also tested whether the lightning GTR-CRISPR system is capable for simultaneous eight-gene disruptions using either one plasmid containing three different markers (A2A2A2A2) or two plasmids (two markers) and each with four gRNAs (A2A2). In both cases, we got either no colonies or very

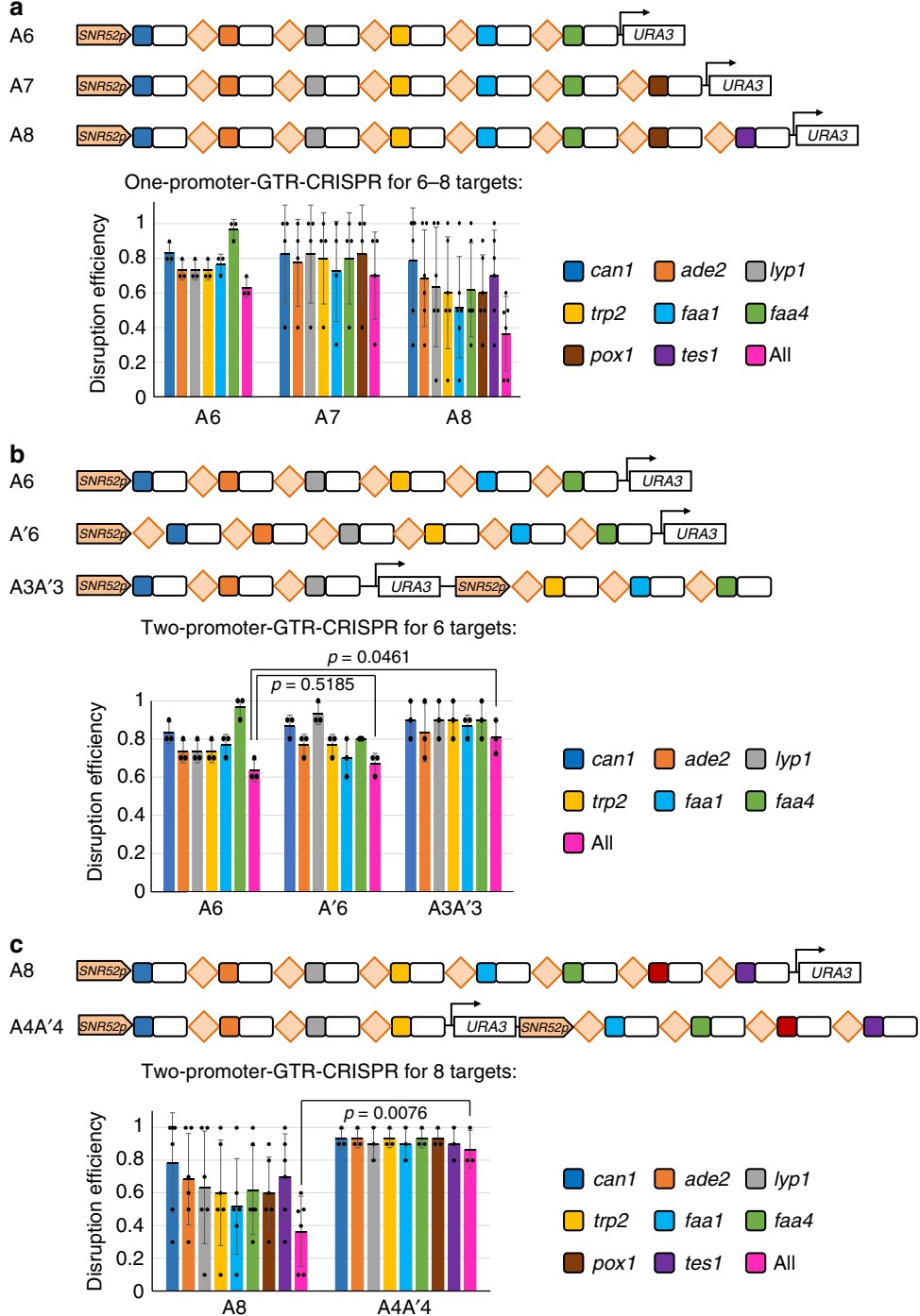

**Fig. 3** Simultaneous disruptions for 6–8 genes using GTR-CRISPR. **a** Graphic representation and results of the one-promoter-GTR-CRISPR system for 6–8 genes. **b** Graphic representation and results of one-fusion-promoter- or two-promoter-GTR-CRISPR system for 6 genes. **c** Graphic representation and results of simultaneous 8-gene disruptions by the two-promoter-GTR-CRISPR system. The data of bar charts represent mean averages of each gene or overall disruption efficiencies. Each black dot represents the gene disruption efficiency of 10 colonies from each biological replicate, and the error bars indicate the standard deviations of all ($n \geq 3$) biological replicates. The statistical analyses are performed using unpaired $t$ Test. Source data are provided as a Source Data file

low efficiency (~1%) of red colonies (*ADE2* disrupted). Thus we concluded that it is not possible to perform efficient lightning GTR-CRISPR for eight targets with current conditions. This could be due to a loop-out issue or low plasmid construction efficiencies with large number of repetitive sequences.

So far, in order to avoid the gRNA efficiency bias, we mainly used reported gRNAs that are shown to be effective for

demonstration of the GTR efficiency. However, in routine experiment, researchers would have less chances using optimal gRNAs. Thus, to further demonstrate our system, we chose 6 *HIS* genes (*HIS1*, *2*, *3*, *5*, *6*, and *7*) for gene disruptions (Fig. 5a). All gRNA targeting sequences were predicted as first hits using the website (link in Methods). The results showed that the gRNA efficiencies indeed have great impacts on gene disruption

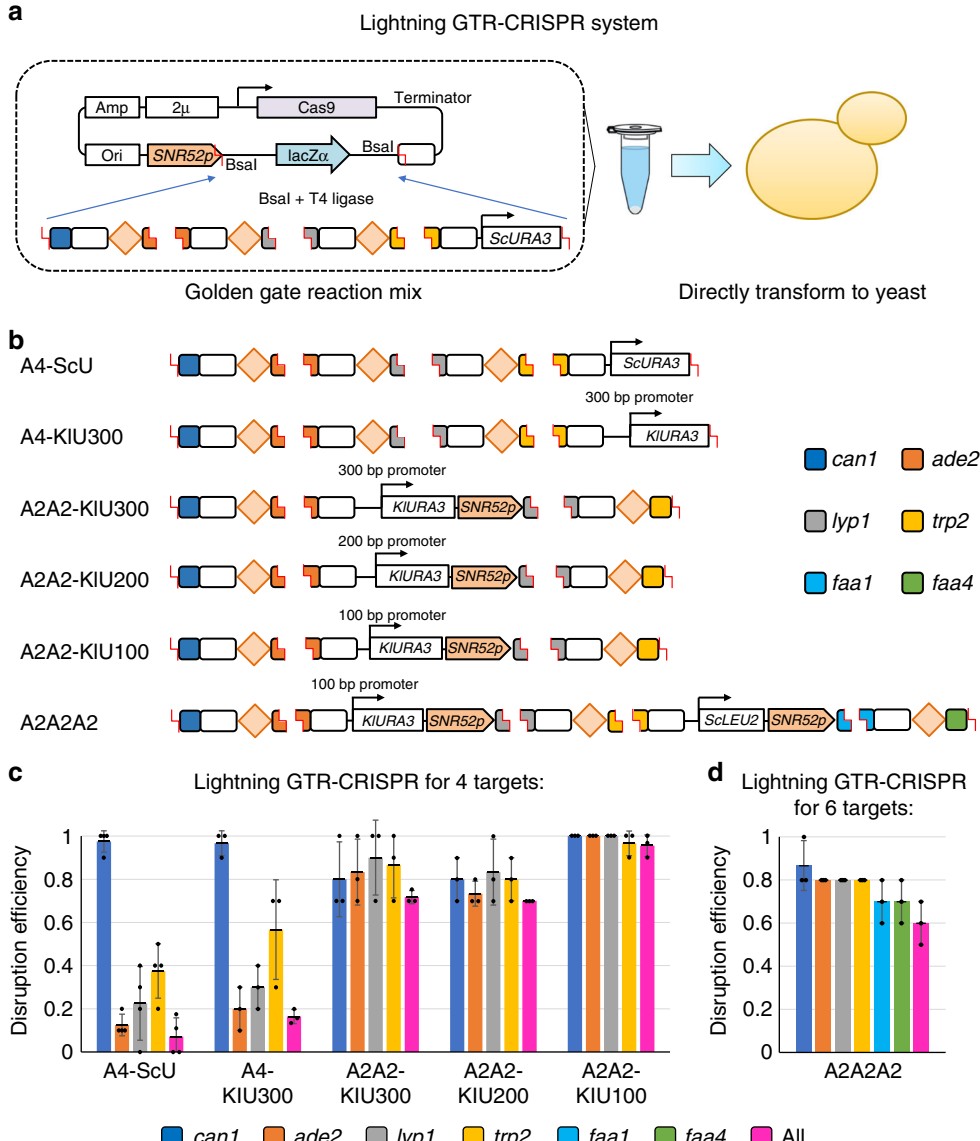

**Fig. 4** Simultaneous disruptions for 4 and 6 genes using Lightning GTR-CRISPR. **a** Graphic representations of the Lightning GTR-CRISPR system without *E. coli* transformation. **b** Optimizations of the Lightning GTR-CRISPR system. ScU stands for *S. cerevisiae URA3* gene. KlU300, KlU200, and KlU100 stand for *K. lactis URA3* gene (*KlURA3*) with its 300, 200, or 100 bp promoters, respectively. *ScLEU2* stands for *S. cerevisiae LEU2* gene, an additional selection marker for 6 targets. A4 represents 4 guide RNAs (gRNAs) in mode A (GTR) format, A2A2 represents total 4 gRNAs in 2 sets of GTR format, and A2A2A2 represents total 6 gRNAs in 3 sets of GTR format. **c**, **d** Results of the Lightning GTR-CRISPR systems for 4-gene and 6-gene disruptions. The data of bar charts represent mean averages of each gene or overall disruption efficiencies. Each black dot represents the gene disruption efficiency of 10 colonies from each biological replicate, and the error bars indicate the standard deviations of all ($n = 3$) biological replicates. Source data are provided as a Source Data file

efficiencies; however, the Lightning GTR-CRISPR was still capable of disrupting 6 genes with a 23.3% efficiency ($n = 6$) in 3 days (Fig. 5b). This result also suggests that the current gRNA design is another limitation besides their co-expression, especially when using our Lighting GTR-CRISPR. Improvements of gRNA design, e.g. identification of the relationship of gRNA sequence and their efficiencies and development of more accurate gRNA design software, may hence boost the performance and application of the Lighting GTR-CRISPR.

**Demonstration of Lightning GTR-CRISPR.** Recently, yeast production of FFAs has attracted much attention for production of biofuels, cosmetics, and pharmaceutical ingredients[22]. In *S. cerevisiae*, fatty acyl chains are accumulated in three major

lipid classes, sterol esters (SEs), triacylglycerols (TAGs), and phospholipids, while FFAs are maintained at low levels in the cytoplasm. Our group has reported that FFA production can be improved through simplifying yeast lipid metabolism by decreasing yeast endogenous production of SEs and TAGs[23]. Following this study and as a demonstration of the general applicability of the Lightning GTR-CRISPR system in *S. cerevisiae*, 8 genes (*FAA1*, *FAA4*, *POX1*, *ARE1*, *ARE2*, *PAH1*, *LPP1*, and *DPP1*) were selected for whole open reading frame (ORF) deletions for FFA production (Fig. 6a).

Whole ORF deletion is sometimes more attractive than gene disruptions (8 bp frameshifting deletions around the PAM sequences), because: (i) the expression of frame-shifted nonsense proteins still consume substrate and energy, which may compete

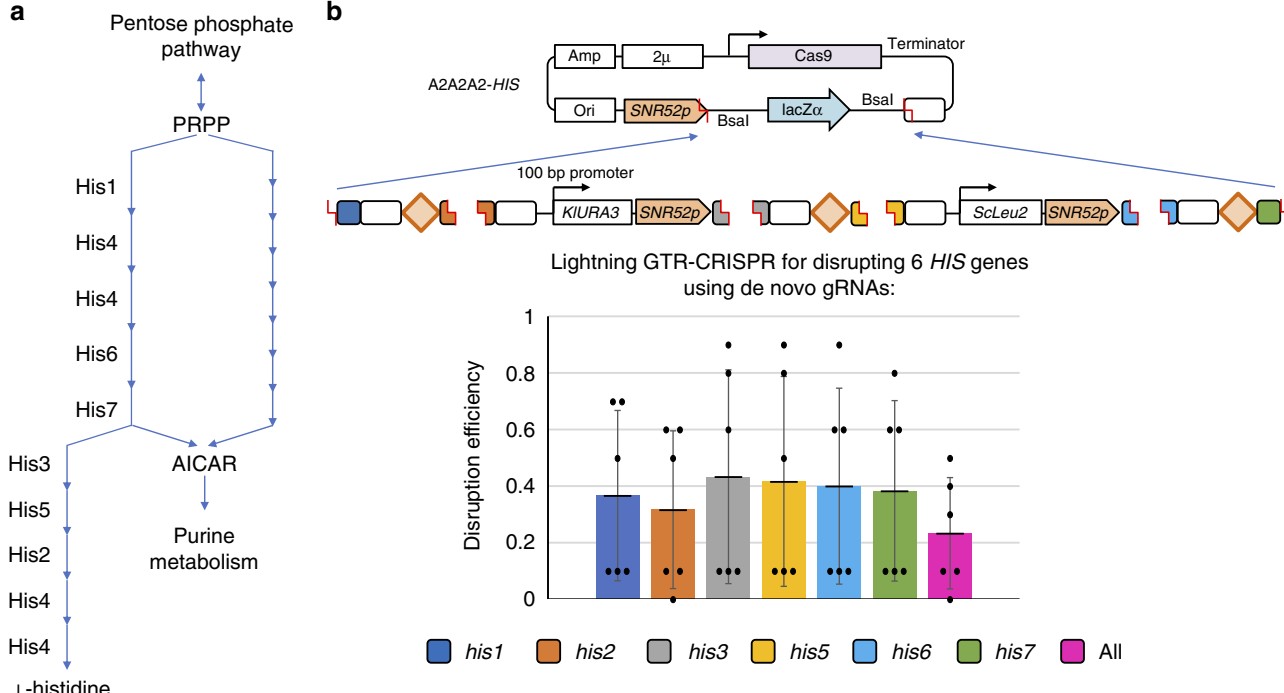

Fig. 5 Lighting GTR-CRISPR with un-optimized guide RNAs. **a** Graphic representations of the L-Histidine synthesis pathway. PRPP phosphoribosyl pyrophosphate, AICAR 5-aminoimidazole-4-carboxamide ribonucleotide. **b** Graphic representations and result of the Lighting GTR-CRISPR systems for six HIS gene disruptions. The data of bar charts represent mean averages of each gene or overall disruption efficiencies. Each black dot represents the gene disruption efficiency of 10 colonies from each biological replicate, and the error bars indicate the standard deviation of all ($n = 6$) biological replicates. Source data are provided as a Source Data file

with cellular resources and course cell stress and (ii) the partially expressed peptides and the frameshifted proteins may have unknown functions. So, we first tested the Lightning GTR-CRISPR for the whole ORF deletion, and to avoid the gene target and gRNA bias, the gRNAs in the GTR-CRISPR 4-gene-disruption experiment were used for deletions. To increase the deletion efficiency, longer donors (120 bp) were used compared with the 100 bp donors used for gene disruptions (Supplementary Figure 3), and an efficiency of simultaneous 4 ORF deletion at 83.3% was obtained in 3 days (Supplementary Figure 4).

Then the Lightning GTR-CRISPR was applied to simplify yeast lipid metabolism through two-round ORF deletions for these eight genes. After the first-round deletion of FAA1, FAA4, POX1, and ARE2, the plasmid was anti-selected by growing the cells on 5-FOA medium, and the second-round deletions of PAH1, LPP1, DPP1, and ARE1 were carried out. The 8-gene deletion GTR3 strain was constructed in 10 days, with FFA (intracellular and extracellular) production reaching 559.52 mg/L in the shake flask, which represents a 30-fold increase compared with the wild-type yeast production of 19.93 mg/L (Fig. 6b and Supplementary Figure 5), demonstrating that the Lightning GTR-CRISPR system can be applied for rapid genome editing. Compared to the highest yield of FFAs reported in S. cerevisiae of about 1.2 g/L in the shake flask[24], the GTR3 does not have any modification on fatty acid synthetic pathway, precursors, or co-factors. Moreover, we reported that removing genes in TAG and SE pathways will increase the production of total fatty acids but not FFAs (Fig. 6b, GTR2). The increase of total fatty acids may be due to the increased phospholipid contents, whereas without disruptions of fatty acid activation pathways, the FFAs may not be accumulated. We also observed that, for both FFAs and total fatty acids, combining the eight knockouts gave higher levels (Fig. 6b, GTR3).

## Discussion

The era of synthetic biology and automation requires genome-editing tools to be multiplexable and highly efficient, as well as with simple procedure and low cost. Here we report the development of a GTR-CRISPR system for multiplexed genome editing in S. cerevisiae. We have demonstrated that using reported gRNAs: (i) The GTR-CRISPR is able to disrupt 8 genes with over 80% efficiencies. (ii) The Lightning GTR-CRISPR can be used for simultaneous 4-gene disruptions at 95.6% and 6-gene disruptions at 60% efficiency without the E. coli transformation step. (iii) Besides the high efficiency reported, both systems significantly reduce the workflow of yeast genome engineering, by eliminating gene synthesis; pre-Cas9 integration; or co-transformation of a separate Cas9 plasmid, helper plasmid construction, and sub-cloning. (iv) We compared two general strategies for gRNA expression and demonstrated that the GTR format is better than individual cassettes when the gRNA number is around 4–5, but two strategies need to be combined for expression of more gRNAs in S. cerevisiae. Thus we expect that the Lightning GTR-CRISPR may be more applicable in automated platforms compared with current genome-editing tools in S. cerevisiae.

Further optimization of GTR-CRISPR systems may include: (i) Identification of stronger RNA polymerase promoters III than the SNR52 promoter and better tRNAs than the tRNA$^{Gly}$. (ii) Increase of the genome-editing efficiency of the GTR-CRISPR system by applying additional SNR52 promoters for simultaneous disruptions of more genes (>8). We stopped our testing at two sets of SNR52 promoter-derived GTR transcripts with a total targeting number of eight, because of limited plasmid construction tools available for assembly of multiple fragments with repetitive sequences (for each target, one gRNA scaffold and one tRNA$^{Gly}$ are required). Thus to optimize current plasmid construction tools for assembly of repetitive sequences might be an

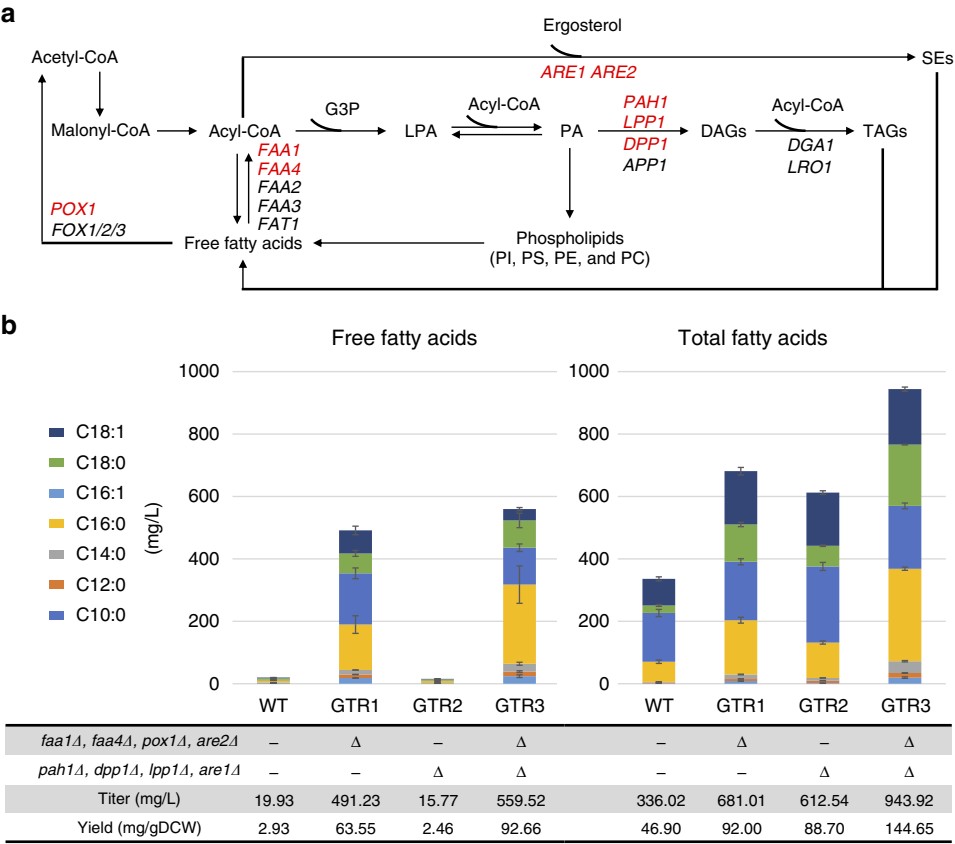

**Fig. 6** Application of the Lightning GTR-CRISPR for free fatty acids production. **a** Schematic representations of lipid metabolism in *S. cerevisiae*. Gene names in red represent deletion targets by the Lightning GTR-CRISPR system. G3P glyceraldehyde 3-phosphate, LPA lysophosphatidic acid, PA phosphatidic acid, DAG diacylglycerol, TAG triacylglycerol, SE sterol ester, PI phosphatidylinositol, PC phosphatidylcholine, PE phosphatidylethanolamine, PS phosphatidylserine. *FAA1* and *FAA4* encode fatty acyl-CoA synthetases; *POX1* encodes a fatty acyl-CoA oxidase; *PAH1*, *DPP1*, and *LPP1* encode phosphate phosphatases, which dephosphorylate PA to yield diacylglycerol; and *ARE1* and *ARE2* encode acyl-CoA:sterol acyltransferases. **b** Free fatty acid and total fatty acid production in yeast deletion strains. Free fatty acids (intracellular and extracellular) and total fatty acids (including fatty acid chains in lipids: TAGs, SEs, phospholipids, etc.) were quantified when the strains were grown for 72 h in minimal medium containing 2% glucose. The data points of bar charts represent mean averages of indicated fatty acids. The error bars indicate the standard deviations of three biological replicates. Source data are provided as a Source Data file

attractive research direction in the future. (iii) Identification of the relationship between gRNA sequences and gRNA efficiencies and development of more reliable gRNA prediction software. (iv) The increase of the genome-editing efficiency of the Lightning GTR-CRISPR system by optimizing the Golden Gate plasmid assembly method. One challenge of the Lightning GTR-CRISPR is that only a small amount was found to be correctly assembled in the Golden Gate reaction mix. This appeared to be a key problem for the Lightning GTR-CRISPR, because when co-transformed into yeast, there are high chances that those unassembled fragments would be circularized through homologous recombination, and either no colonies or colonies with misassembled plasmids would be found. To further increase targeting numbers and efficiencies of Lightning GTR-CRISPR system, optimizing the Golden Gate assembly method to decrease the percentage of un-assembled fragments might be needed.

In conclusion, GTR-CRISPR system will be an invaluable addition to the toolbox of synthetic biology of *S. cerevisiae*, as well as shed lights on CRISPR-based multiplexing in other non-model organisms.

## Methods

**Strains and media**. The yeast strain used in this study was CEN.PK 113-5D (*MATa*, *MAL2-8c*, *SUC2*, *ura3-52*) and CEN.PK 113-5D *leu2Δ* (*MATa*, *MAL2-8c*, *SUC2*, *ura3-* 52, *leu2Δ*). Strains were grown in YPD media with 2% glucose before transformation. The transformants were plated on synthetic complete (SC) media minus the auxotrophic compound complemented by the plasmids. SC-URA agar plates were used to select GTR-CRISPR-disrupted cells, and SC-URA-LEU agar plates were used to select cells with Lightning GTR-CRISPR-based disruptions of six genes.

**Design of gRNA targeting sequences**. The gRNAs for *CAN1*, *ADE2*, *LYP1*, *FAA1*, *FAA4*, *POX1*, and *TES1* were selected from published papers[8,14]. The gRNAs for *TRP2*, *ARE1*, *ARE2*, *PAH1*, *LPP1*, *DPP1*, *HIS1, 2, 3, 5, 6*, and *7* were predicted by the website: https://atum.bio/eCommerce/cas9/input. All gRNA sequences are listed in Supplementary data 1.

**Plasmid construction**. A high copy number 2μ-based plasmid backbone with a mutated Cas9 (D147Y and P411T) was adapted from the reported HI-CRISPR plasmid[1]. The pCas vector was constructed with the lacZα sequence flanked by an *SNR52* promoter and a gRNA scaffold with BsaI cleavage site at the end of the *SNR52* promoter (GATC) and the beginning of the gRNA scaffold (GTTT). To assemble multiple gRNAs on one plasmid, the lacZα sequence is removed by BsaI digestion and replaced by PCR-generated fragments (Fig. 1). The PCR templates were plasmids containing gRNA scaffold with a tRNA sequence or gRNA scaffold with a selection marker. For the first and last gRNAs, the 20 bp gRNA targeting sequences were designed all on the primers and these two sites could be ligated on pCas vector. The 4-bp sequences of other 20 bp gRNA targeting sequences can be used as Golden-Gate ligation sites to assemble different fragments. All primers were designed with no more than 60 bp. The sequences of oligonucleotide primers for generating donors and plasmids are listed in Supplementary data 1. The DNA sequences of PCR templates and pCas plasmid are provided in Supplementary Note. As a showcase, the design for Lightning

GTR-CRISPR for four-gene disruptions is described in Supplementary Figure 1 and Supplementary Table 1. For a 20 μL total Golden gate reaction, 2 μL 10 × T4 Ligase buffer (M0202, New England Biolabs), 1.6 μL BsaI (R0535, New England Biolabs), 0.4 μL T4 Ligase (M0202, New England Biolabs), 150 ng for pCas plasmid (8.7 kb), and other fragments with a molar 1:1 ratio with the pCas were added into the reaction mix. The Golden gate reaction was carried out using the following temperature profile: Step 1, 37 °C for 30 min; Step 2, 37 °C for 10 min; Step 3, 16 °C for 5 min; Step 4, repeat steps 2 and 3 for 16 cycles; Step 5, 16 °C for 30 min; Step 6: 37 °C for 30 min, Step 7, 80 °C for 6 min; Step 8, 4 °C hold. For the lightning GTR-CRISPR disrupting 6 genes, 30 cycles were used instead of 16 cycles.

**Preparation of repairing dsDNA (donors).** This method is developed from the previous publication[25]. The oligonucleotide primers are synthesized in the GENEWIZ company using standard desalting purification method, and diluted at 100 mM. Then 10 μL of 5 × Q5 reaction buffer, 10 μL of each of the two oligonucleotide primers, 10 μL of 10 mM dNTPs, 1 μL of Q5 enzyme, and 9 μL of ddH₂O are used for a 50-μL PCR reaction. The PCR reaction is set up by 98 °C for 10 s, 58 °C for 20 s, and 72 °C for 20 s, a total of 35 cycles. The PCR products are purified by ethanol precipitation and diluted by ddH₂O at concentration 10 μg/μL.

**Yeast transformation.** Yeast transformation was carried out using the electroporation. Inoculate the overnight precultured cells from a single colony into 50 ml YPD media at an initial OD600 of 0.3 and then culture at 30 °C for around 5 h, until OD600 reaches approximately 1.6. The yeast cells were collected by centrifugation at 3000 rpm for 3 min at 4 °C and washed once with 20 mL ice-cold 1 M sorbitol. The yeast cells were resuspended in 16 ml 1 M sorbitol, 2 mL 10 × TE (100 mM Tris-HCL, 10 mM EDTA, pH 7.5), and 2 mL 1 M lithium acetate and the cultures were incubated in the shaker for 30 min at 30 °C. The cells were washed twice with 20 mL ice-cold sorbitol. All the liquid was removed and cells were resuspended in 0.4 mL 1 M sorbitol. The competent cells could be stored at −80 °C before use. In all, 16.5 μg donors with 0.5 μg constructed GTR-CRISPR plasmid or 10 μL Golden Gate reaction mix (not purified for 4 targets) for the Lightning GTR-CRISPR system were added to 100 μL yeast competent cells in prechilled electroporation cuvette (0.2 cm). For Lightning GTR-CRISPR system of 6 targets, 80 μL Golden Gate reaction products was purified by PCR purification column and all purified DNA was transformed to yeast for each transformation. After the electroporation, yeast cells were recovered in 2 mL of YPD medium with 1 M Sorbitol (1:1) for 5 h and transferred to SC media minus the auxotrophic compound complemented by the plasmids for 24 h, then plated on the same SC drop-out plate.

**Calculation of gene-disruption efficiency.** To evaluate disruption efficiencies, ten colonies from each biological replicate (at least three biological replicates) were randomly selected and streaked on different synthetic media plates minus the auxotrophic components or with additive supplements. SC-URA-ARG (SC-uracil and arginine) with 60 mg/L L-canavanine (Sigma C1625) agar plates were used to select for CAN1 disrupted cells. SC-URA-ADE (SC-uracil and adenine) agar plates were used to select for ADE2 disrupted cells. SC-URA-LYS (SC-uracil and lysine) with 250 mg/L thialysine (S-2-aminoethyl-L-cysteine, Sigma A2636) plates were used to select for LYP1 disrupted cells. SC-TRP (SC-tryptophan) plates were used to select for TRP2 disrupted cells. All disrupted targets without phenotypes were verified by PCR amplification of disruption sites using upstream and downstream ~200 bp primers and followed by XbaI digestion (these donors contain an XbaI site in middle in place of 8 bp around PAM sequence). The ORF deletions were verified by PCR amplification using ORF upstream and downstream ~200 bp primers. The mean averages and standard deviations of all ($n \geq 3$) biological replicates were calculated. The statistical analyses were performed using unpaired t Test: p values were calculated with two sides/tails and hypothesis of two sample assuming equal variances.

**Quantification of FFA and total fatty acid content.** Samples for FFAs and total fatty acids were taken in the shake flask cultures at 72 h, extracted, and derivatized to fatty acyl methyl esters for quantitation through gas chromatography–mass spectrometry, as stated in our previous papers[23].

**Reporting summary.** Further information on experimental design is available in the Nature Research Reporting Summary linked to this article.

## Data availability

The source data underlying Figs. 2b–d, 3a–c, 4c, d, 5b, and 6b and Supplementary Figs. 2b and 4 are provided as a Source Data file.

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

## Acknowledgements

This work was supported by the Beijing Advanced Innovation Center for Soft Matter Science and Engineering, Beijing University of Chemical Technology, the Beijing Municipal Natural Science Foundation (5182017), the Fundamental Research Funds for the Central Universities (buctrc201801), State Key Laboratory of Microbial Technology Open Projects Fund (Project NO. M2017-06), State Key Laboratory of Chemical

Resource Engineering, the International Cooperation Program of Beijing Municipal Science & Technology Commission, the Novo Nordisk Foundation (Grant NO. NNF10CC1016517), and the Knut and Alice Wallenberg Foundation. We also acknowledge Professor Huimin Zhao from the University of Illinois at Urbana-Champaign for kindly providing the pCRCT plasmid and Raphael Ferreira from the Chalmers University of Technology for discussions.

## Author contributions

Yueping Zhang, J.W., J.N., and Z.L. designed the research; Yueping Zhang., J.W., Z.W., and Yiming Zhang carried out the experiment; Yueping Zhang, J.W., S.S., J.N., and Z.L. analysed data; Yueping Zhang, J.N., and Z.L. wrote the paper. J.N. and Z.L. supervised the research.

## Additional information

**Competing interests:** The authors declare no competing interests.

