## [Peer Review File · Nature Communications]

Reviewers' Comments:

Reviewer #1:

In this manuscript, Zhang et al. report on an improved multiplexed CRISPR-Cas system for *S. cerevisiae*. By optimizing gRNA expression/processing they demonstrate the knockout of up to 8 genes (using previously reported gRNAs). The science appears sound, the conclusions and discussion mostly appropriate (see comments below).

Regarding conceptual novelty, this work represents a step forward compared to previous efforts. So far multiplexed CRISPR-Cas systems have already been reported for *S. cerevisiae* using similar concepts and the authors themselves also discuss these efforts ("*Currently, the gRNA processing capability of yeast *Saccharomyces cerevisiae* is reported to be up to five gRNAs in one construct with 50%-100% genome editing efficiencies*" [...] "*similar concepts have been reported for targeting 1-3 genome loci, however, such methods often suffer from low efficiencies or non-equimolar expression of gRNAs*"). Hence, this paper appears as a thorough, systematic optimization of state of the art technologies (pushing the number of gRNAs in one construct from 5 to 8, while maintaining similar efficiencies).

These optimized strategies will surely be useful for researchers working on multiplexed CRISPR-Cas genome editing in *S. cerevisiae* and other yeast species, possibly even beyond yeasts.

The manuscript contains also an interesting approach of directly transforming Golden Gate assemblies, here the authors provide deeper insights into issues hampering these efforts in *S. cerevisiae*. Due to strong HR it appears that identical sequences are looped out, and the authors provide a smart solution to this issue by placing the marker cassette in the middle of their repetitive assembly.

Overall, the issues listed below should be resolved and clarifications provided. Eventually publication will mostly depend on the level of novelty required by this journal and the respective editorial policies.

Major issues

1. The results reported here represent a best case scenario: The vast majority of gene disruption targets and gRNA sequences used here were selected from published papers to avoid gRNA efficiency bias. In real world applications, researchers would not have access to such optimal gRNAs when targeting new/unprecedented genes. These issues should at least be mentioned and adequately discussed in the discussion section. Ideally the authors should perform additional experiments with de novo gRNAs to provide an estimate for the real world applicability of their system. An experiment close to real world conditions, would be to pick target genes, that have not been reported in the literature, or to design new gRNAs against the target genes in this study using standard tools reported in the literature
2. The main text does not explain how the building blocks for the gRNA/tRNA fusion parts are generated (ordered as synthetic double stranded DNA fragments? or generated from oligoes by annealing or PCR?). This information should be provided at least briefly in the course of the main text. Especially since the discussion on the generation of the double stranded donor fragments is quite thorough. Also the description in the M&Ms is surprisingly short: "To assemble multiple gRNAs on one plasmid, the lacZ α sequence is removed by BsaI digestion and replaced by PCR-generated fragments (Fig. 1a)." Again, how are the gRNA-tRNA fusions exactly generated by PCR? By ordering long oligoes? Or by overlapping oligoes amplification as for the donor fragments? What is the cost

associated with that? That is described for the donor fragments but not the actual gRNA array. Explain in detail.

3. I am not sure if I agree with the description “cloning-free”. Even in the lighting GTR setup, it is still necessary to perform a restriction digestion and ligation (Golden Gate style), only the propagation in *E. coli* is omitted. “cloning free” has for me (at least in *S. cerevisiae*) typically the connotation of direct transformation of DNA and relying on the *S. cerevisiae* HR machinery to assemble the cassette. Hence change the wording and rather refer to the method as something like “*E. coli* transformation – free” and mention in the abstract, that the *S. cerevisiae* cells are transformed with a Golden Gate ligation mix.
4. Some key information on statistics is missing:
 - 4.1. While the type of error bars are mentioned in the captions, it is left unclear if mean or median are shown. This information must be added.
 - 4.2. In some figure standard error (Fig. 2, Fig. 3, Fig. 5) shown and in others standard deviation (Fig. 4). Why is the standard error (SE) shown in some cases and not standard deviation (SD) for all of them? I would think that readers would be interested in seeing the dispersion and variability of the data represented (SD). Or is the SE needed to test differences between means? These issues (choice of SE, SD) should be briefly outlined in the M&Ms section, add a section e.g., statistical analyses, and explain the choices for the figures.
 - 4.3. Why do the replicates always group in intervals of 0.1? Because there were always exactly 10 colonies tested? I found this explanation only later in the M&M section “10 colonies from each biological replicate (at least 3 biological replicates)” this information should be added to the captions.
 - 4.4. I am not an expert for statistical analyses, but some of the data presentation should be evaluated (also if the unpaired t-Test for the two-tailed P values is the most meaningful choice, I am not an expert on this). Probably other reviewers have already done so, otherwise possibly the Nat. comm. editorial team has a statistical expert who could have a look?

Minor comments

1. What happens if the 8 gRNA / knockout gene combination tested with lighting in vivo cloning? Or is this just not possible due to the loop out issues?
2. See the attached pdf for a few additional minor comments on the manuscript/wording.

Reviewer #2:

Remarks to the Author:

The manuscript describes the development of new CRISPR systems for multiplexed genome engineering in the yeast *Saccharomyces cerevisiae*. The research employs gRNA-tRNA arrays that have also been shown to be efficient for this purpose in other organisms. The authors then develop a rapid cloning-free system that skips transformation and checking in *E. coli*. While the general approaches are not novel, their adaptation for the rapid knockout of multiple genes in yeast is carefully executed and optimized, and the results obtained are impressive: 8 genes (with 87% efficiency) in 7 days, and 6 genes (with 60% efficiency) with the cloning-free system in 3 days. These are significant improvements over the currently published multiplexing approaches for *S. cerevisiae*. The methods and results will be of interest to a wide range of researchers engineering the *S. cerevisiae* genome, and to those interested in similar tools for other yeast species.

The authors should address the following comments:

1. gRNA-tRNA arrays have been used yeast and in other organisms. The authors should compare their

approach to the literature and point out any important or novel aspects of their system.

2. Figure 1: Panels (b) and (c) could be removed. They provide no information and are rather simplistic for the knowledge of the expected reader.

3. Figure 4 may be confusing as the 4th gRNA (in the A2-A2 constructs) and 5th gRNA (in the A2X3 construct) is not at first apparent. The authors might add some text to the caption pointing out the additional scaffold element on the backbone plasmid.

4. The authors state in the Abstract and Introduction that implementation of their system resulted in a 31-fold increase in fatty acid synthesis. As these gene knockouts have been previously studied, the authors should mention that the promise of their system was verified using 8 previously identified gene knockouts that are known to increase free fatty acid levels (not only in the Results/Discussion).

5. The initial and final fatty acid concentrations (g/L) should be stated in addition to mg/gCDW so that the reader knows the titer achieved. The fold increase alone is not too informative as native free fatty acids are extremely low in *S. cerevisiae*. The authors should also state whether both intracellular and extracellular levels were measured.

6. The synergy observed for the free fatty acid results for GTR3 (both sets of knockouts) should be commented on.

7. Lines 219-222: This is not a significant advance. While some methods use longer primers, many labs reduce costs by using two shorter primers (e.g., we have for 3 years now). The authors should reword to simply state that using 2 primers reduces cost, not indicate that this is unusual/novel.

8. Line169: The authors should comment on their rationale for the selected *K. lactis* URA3 promoter sizes. A comparison of expression with these promoters could be included in the Supplementary Data.

9. The manuscript is generally well-written and easy to follow. However, there are multiple grammatical errors and typos throughout (e.g., lines 32-33, 41, 65, 234, 242/243, 245, etc.) that should be corrected.

Thanks a lot for the reviews of our paper. We have now revised the paper based on the very constructive comments of the reviewers. We have also added the extra experiments requested by two of the reviewers. We feel confident that all issues have been addressed and hope that you will find the revised version of our paper improved. All changes in the paper are highlighted in yellow and our response to the reviewers is given below.

Response to Reviewers:

Reviewer #1 (Remarks to the Author):

In this manuscript, Zhang et al. report on an improved multiplexed CRISPR-Cas system for *S. cerevisiae*. By optimizing gRNA expression/processing they demonstrate the knockout of up to 8 genes (using previously reported gRNAs). The science appears sound, the conclusions and discussion mostly appropriate (see comments below).

Regarding conceptual novelty, this work represents a step forward compared to previous efforts. So far multiplexed CRISPR-Cas systems have already been reported for *S. cerevisiae* using similar concepts and the authors themselves also discuss these efforts (*“Currently, the gRNA processing capability of yeast Saccharomyces cerevisiae is reported to be up to five gRNAs in one construct with 50%-100% genome editing efficiencies” [...] “similar concepts have been reported for targeting 1-3 genome loci, however, such methods often suffer from low efficiencies or non-equimolar expression of gRNAs”*). Hence, this paper appears as a thorough, systematic optimization of state of the art technologies (pushing the number of gRNAs in one construct from 5 to 8, while maintaining similar efficiencies).

These optimized strategies will surely be useful for researchers working on multiplexed CRISPR-Cas genome editing in *S. cerevisiae* and other yeast species, possibly even beyond yeasts.

The manuscript contains also an interesting approach of directly transforming Golden Gate assemblies, here the authors provide deeper insights into issues hampering these efforts in *S. cerevisiae*. Due to strong HR it appears that identical sequences are looped out, and the authors provide a smart solution to this issue by placing the marker cassette in the middle of their repetitive assembly.

Overall, the issues listed below should be resolved and clarifications provided. Eventually publication will mostly depend on the level of novelty required by this journal and the respective editorial policies.

Major issues

1. The results reported here represent a best case scenario: The vast majority of gene disruption targets and gRNA sequences used here were selected from published papers to avoid gRNA efficiency bias. In real world applications, researchers would not have access to such optimal gRNAs when targeting new/unprecedented genes. These issues should at least be mentioned and adequately discussed in the discussion section. Ideally the authors should perform additional experiments with de novo gRNAs to provide an estimate for the real world applicability of their system. An experiment close to real world conditions, would be to pick target genes, that have not been reported in the literature, or to design new gRNAs against the target genes in this study using standard tools reported in the literature

We thank for this constructive criticism and we agree with the reviewer that we need a real

application for our Lightning GTR-CRISPR system. We therefore chose 6 *HIS* genes (*HIS1*, 2, 3, 5, 6, and 7) for an additional demonstration of our method. All gRNAs were predicted as first hits using website (<https://atum.bio/eCommerce/cas9/input>). The simultaneous disruption efficiency is 23.3% (n=6) using A2A2A2 Lightning GTR-CRISPR system. (Supplementary Fig. S6).

2. The main text does not explain how the building blocks for the gRNA/tRNA fusion parts are generated (ordered as synthetic double stranded DNA fragments? or generated from oligoes by annealing or PCR?). This information should be provided at least briefly in the course of the main text. Especially since the discussion on the generation of the double stranded donor fragments is quite thorough. Also the description in the M&Ms is surprisingly short: "To assemble multiple gRNAs on one plasmid, the lacZ α sequence is removed by BsaI digestion and replaced by PCR-generated fragments (Fig. 1a)." Again, how are the gRNA-tRNA fusions exactly generated by PCR? By ordering long oligoes? Or by overlapping oligoes amplification as for the donor fragments? What is the cost associated with that? That is described for the donor fragments but not the actual gRNA array. Explain in detail.

We thank the review for this suggestion and we have now added all the required descriptions in the Plasmid Construction part of the Methods and there is an example of primer design in Supplementary Fig. S2 and Supplementary Table S1.

3. I am not sure if I agree with the description "cloning-free". Even in the lightning GTR setup, it is still necessary to perform a restriction digestion and ligation (Golden Gate style), only the propagation in *E. coli* is omitted. "cloning free" has for me (at least in *S. cerevisiae*) typically the connotation of direct transformation of DNA and relying on the *S. cerevisiae* HR machinery to assemble the cassette. Hence change the wording and rather refer to the method as something like "*E. coli* transformation-free" and mention in the abstract, that the *S. cerevisiae* cells are transformed with a Golden Gate ligation mix.

We have deleted all terms of "cloning-free" or changed it to "*E. coli* transformation-free" in the paper.

4. Some key information on statistics is missing:

4.1. While the type of error bars are mentioned in the captions, it is left unclear if mean or median are shown. This information must be added.

We are sorry for the confusion. The data represents mean averages and the block dots represent all biological replicates. This part has been revised in the caption.

4.2. In some figure standard error (Fig. 2, Fig. 3, Fig. 5) shown and in others standard deviation (Fig. 4). Why is the standard error (SE) shown in some cases and not standard deviation (SD) for all of them? I would think that readers would be interested in seeing the dispersion and variability of the data represented (SD). Or is the SE needed to test differences between means? These issues (choice of SE, SD) should be briefly outlined in the

M&Ms section, add a section e.g., statistical analyses, and explain the choices for the figures.

All analysis has been changed to standard deviation (SD). And the statements have been added in the caption.

4.3. Why do the replicates always group in intervals of 0.1? Because there were always exactly 10 colonies tested? I found this explanation only later in the M&M section “10 colonies from each biological replicate (at least 3 biological replicates)” this information should be added to the captions.

Fixed.

4.4. I am not an expert for statistical analyses, but some of the data presentation should be evaluated (also if the unpaired t-Test for the two-tailed P values is the most meaningful choice, I am not an expert on this). Probably other reviewers have already done so, otherwise possibly the Nat. comm. editorial team has a statistical expert who could have a look?

We have changed the statements to “The statistical analyses were performed using unpaired t-Test.” The statistical analyses were performed using two sample unpaired t-Test (Student’s t-Test). The p value is calculated through GraphPad website (<https://www.graphpad.com/quickcalcs/ttest1.cfm>).

Minor comments

1. What happens if the 8 gRNA / knockout gene combination tested with lighting in vivo cloning? Or is this just not possible due to the loop out issues?

We have tested 8 gRNAs with the lightning GTR-CRISPR system using either one plasmid with 3 different markers or two plasmids with each having 4 gRNAs. In both cases, we got either no colonies or very low efficiency (~1%) of red colonies (*ADE2* disrupted). Thus, we concluded that it is not possible to perform efficient lightning GTR-CRISPR for 8 targets with the current conditions. This can be due to a loop-out issue or unsuccessful Gold-Gate reaction (because of too many fragments). We have included a comment about this in the main text “The further extension of Lightning GTR-CRISPR system for 8 gRNAs using a similar strategy was not successful”.

2. See the attached pdf for a few additional minor comments on the manuscript/wording.

Could you cite a reference for this? If not, omit or rewrite. One could also argue the other way: This system is very inefficient and that’s why *S. cerevisiae* needs multiple copies to compensate for this inefficiency. If it were very efficient, a single copy might be enough. Comments for “ (ii) tRNAGly is abundant in yeast with 16 gene copies predicted in *S. cerevisiae*19, which means the native tRNAGly processing system is likely to be very efficient”

We do not have a reference for this, and we have deleted this statement.

Why exactly 8 bp? and not something shorter but 6 or 3? Comments for “we first tested three different modes for introducing simultaneous gene disruptions (8bp frameshifting deletions around PAM sequences) of 3-5 genes in *S. cerevisiae*.”

In this design, we considered that there are some reports of the potential off-target cleavage still occurring with 3-5 bp mismatches in the PAM-distal part of the gRNA sequence (Pattanayak *et al* Nat Biotechnol. 2013 and Hsu *et al* Nat Biotechnol. 2013). Thus, we deleted 8bp including the PAM sequencing to avoid potential off-targets. And we totally agree with the reviewer that the frameshifting deletion can be shorter than 8bp and we have successfully generated shorter deletions using GTR-CRISPR system in other cases.

Reviewer #2 (Remarks to the Author):

The manuscript describes the development of new CRISPR systems for multiplexed genome engineering in the yeast *Saccharomyces cerevisiae*. The research employs gRNA-tRNA arrays that have also been shown to be efficient for this purpose in other organisms. The authors then develop a rapid cloning-free system that skips transformation and checking in *E. coli*. While the general approaches are not novel, their adaptation for the rapid knockout of multiple genes in yeast is carefully executed and optimized, and the results obtained are impressive: 8 genes (with 87% efficiency) in 7 days, and 6 genes (with 60% efficiency) with the cloning-free system in 3 days. These are significant improvements over the currently published multiplexing approaches for *S. cerevisiae*. The methods and results will be of interest to a wide range of researchers engineering the *S. cerevisiae* genome, and to those interested in similar tools for other yeast species.

The authors should address the following comments:

1. gRNA-tRNA arrays have been used yeast and in other organisms. The authors should compare their approach to the literature and point out any important or novel aspects of their system.

We now compare our approach with other approaches described in the literature in the discussion section.

2. Figure 1: Panels (b) and (c) could be removed. They provide no information and are rather simplistic for the knowledge of the expected reader.

Fixed.

3. Figure 4 may be confusing as the 4th gRNA (in the A2-A2 constructs) and 5th gRNA (in the A2X3 construct) is not at first apparent. The authors might add some text to the caption

pointing out the additional scaffold element on the backbone plasmid.

We changed the term from "A2x3" to "A2A2A2" to make it more clearly. We also added text in the caption pointing out the additional scaffold elements in figure 4.

4. The authors state in the Abstract and Introduction that implementation of their system resulted in a 31-fold increase in fatty acid synthesis. As these gene knockouts have been previously studied, the authors should mention that the promise of their system was verified using 8 previously identified gene knockouts that are known to increase free fatty acid levels (not only in the Results/Discussion).

We included the "previously identified eight genes" in the abstract and introduction.

5. The initial and final fatty acid concentrations (g/L) should be stated in addition to mg/gCDW so that the reader knows the titer achieved. The fold increase alone is not too informative as native free fatty acids are extremely low in *S. cerevisiae*. The authors should also state whether both intracellular and extracellular levels were measured.

We included both "mg/L" and "mg/gCDW" in the Figure 5. We have included the description of "Free fatty acids (intracellular and extracellular) and total fatty acids (including fatty acid chains in lipids: TAGs, SEs, Phospholipids, and etc.)" in the caption of figure 5 and also in the main text.

6. The synergy observed for the free fatty acid results for GTR3 (both sets of knockouts) should be commented on.

Fixed.

7. Lines 219-222: This is not a significant advance. While some methods use longer primers, many labs reduce costs by using two shorter primers (e.g., we have for 3 years now). The authors should reword to simply state that using 2 primers reduces cost, not indicate that this is unusual/novel.

We have deleted this statement.

8. Line169: The authors should comment on their rationale for the selected *K. lactis* URA3 promoter sizes. A comparison of expression with these promoters could be included in the Supplementary Data.

We compared the gene expression of these promoters and included the data in Supplementary Fig. S5.

9. The manuscript is generally well-written and easy to follow. However, there are multiple grammatical errors and typos throughout (e.g., lines 32-33, 41, 65, 234, 242/243, 245, etc.)

that should be corrected.

Thanks. We have corrected all indicated and other grammatical errors.

Reviewers' Comments:

Reviewer #1:

The authors have responded diligently to my comments and provide additional data demonstrating the performance of their system for de novo gRNA design targeting multiple genes ("real world" example).

If a few minor comments are addressed (see below), I feel the manuscript could be ready for publication depending on the level of novelty required by this journal and the respective editorial policies.

1. About the new "real world" HIS example provided by the authors (their rebuttal:

We thank for this constructive criticism and we agree with the reviewer that we need a real application for our Lightning GTR-CRISPR system. We therefore chose 6 *HIS* genes (*HIS1*, 2, 3, 5, 6, and 7) for an additional demonstration of our method. All gRNAs were predicted as first hits using website (<https://atum.bio/eCommerce/cas9/input>). The simultaneous disruption efficiency is 23.3% (n=6) using A2A2A Lightning GTR-CRISPR system. (Supplementary Fig. S6.)

This information should not be hidden in the supporting information, move it to the main manuscript. I believe Nat. comm. allows for up to 10 display items, so this should be easily possible.

Furthermore, this information should also be added to the abstract. The authors have currently specified in the abstract the efficiency of 87% for 8 genes "Using defined gRNAs reported earlier". Add a sentence that mentions the knockout of 6 genes using de novo designed gRNAs with 23% efficiency.

Thereby readers are immediately provided with the key information.

Also, it might be good to add a conclusion/summary statement on the implications of these results in the manuscript – ultimately this suggests that your system works quite well and there is rather an issue with our ability to generate efficient gRNAs against unprecedented targets. Something like "With previously functionally verified gRNAs, we could knock out up to 8 genes at 87% efficiency. Using de novo designed gRNAs, the efficiency reduced to 23% for 6 genes, whereas 8 genes could not be successfully knocked out. These results may suggest, that current gRNA design is rather a limitation than their co-expression using our lightning system. Improvements of gRNA design may hence also boost the performance of our lightning system in future applications."

Update: I saw later on that the authors briefly mentioned this in the discussion section. I think it would still be good to mention the explicit results, how to come to this conclusion.

2. About my previous comment "What happens if the 8 gRNA / knockout gene combination tested with lightning in vivo cloning? Or is this just not possible due to the loop out issues?"

The authors replied:

"We have tested 8 gRNAs with the lightning GTR-CRISPR system using either one plasmid with 3 different markers or two plasmids with each having 4 gRNAs. In both cases, we got either no colonies or very low efficiency (~1%) of red colonies (*ADE2* disrupted). Thus, we concluded that it is not possible to perform efficient lightning GTR-CRISPR for 8 targets with the current conditions. This can be due to a loop-out issue or unsuccessful Gold-Gate reaction (because of too many fragments). We have included a comment about this in the main text "The further extension of Lightning GTR-CRISPR system for 8 gRNAs using a similar strategy was not successful"."

This explanation is sound and should be provided as part of the manuscript, I imagine this could be quite relevant to some readers. So extend the one sentence you have already written with the full explanation you have written to me (add a “data not shown” for the 1% red colonies etc.).

3. About my previous comment “Why exactly 8 bp? and not something shorter but 6 or 3? Comments for “we first tested three different modes for introducing simultaneous gene disruptions (8bp frameshifting deletions around PAM sequences) of 3-5 genes in *S. cerevisiae*.”

Authors’ reply:

In this design, we considered that there are some reports of the potential off-target cleavage still occurring with 3-5 bp mismatches in the PAM-distal part of the gRNA sequence (Pattanayak *et al*/ Nat Biotechnol. 2013 and Hsu *et al*/ Nat Biotechnol. 2013). Thus, we deleted 8bp including the PAM sequencing to avoid potential off-targets. And we totally agree with the reviewer that the frameshifting deletion can be shorter than 8bp and we have successfully generated shorter deletions using GTR-CRISPR system in other cases.

This is interesting and relevant information, please also add these references and the brief explanation to the manuscript.

4. The authors have only marked changes to the previous version with yellow highlighting. Thereby the initial text has been lost. Or it is required to tediously compare the two versions. I am not willing to do that, so my review is somewhat incomplete and the editorial teams needs to decide if they want to manually check the changes.

For any future resubmissions, the authors should use the “Track changes” functionality of Word, allowing to compare new and old text.

Reviewer #2:

Remarks to the Author:

Summary:

The manuscript describes the development of new CRISPR systems for multiplexed genome engineering in the yeast *Saccharomyces cerevisiae*. The research employs gRNA-tRNA arrays that have also been shown to be efficient for this purpose in other organisms. The authors then develop a rapid cloning-free system that skips transformation and checking in *E. coli*. While the general approaches are not novel, their adaptation for the rapid knockout of multiple genes in yeast is carefully executed and optimized, and the results obtained are impressive. These are significant improvements over the currently published multiplexing approaches for *S. cerevisiae*. The methods and results will be of interest to a wide range of researchers engineering the *S. cerevisiae* genome, and to those interested in similar tools for other yeast species.

Re-review Comments:

The authors have addressed the comments from the reviewers, and their response and revisions are generally satisfactory. They have included missing details, added new experimental results, and have improved the manuscript. However, a remaining concern is in the preparation – there are still significant grammatical/wording issues, including in the new text added to the manuscript:

1. In response to a reviewer comment, the authors now refer to their gRNAs as “defined gRNAs”. However, it is unclear what is meant by “defined”. The authors should explicitly state that these were optimal gRNAs previously shown to be effective, and then refer to them simply as gRNAs.

2. New paragraph on p. 8: This needs to be carefully rewritten as there are several errors and some sentences are unclear. Two examples: “unprecedented” appears to be an incorrect word choice, and line 85: “system was still enabled simultaneous 6 gene disruptions” needs to be fixed. Also, as mentioned above, “defined gRNA” is unclear and needs to be described more completely.

3. Other examples of corrections needed:

- Abstract, lines 31-34: suggest changing to “by about 30-fold in 10 days using just two rounds of deletions of eight previously identified genes”
- Line 54: change “maximum” to “a maximum”
- Lines 62-63: remove either “Therefore” or “thus” (redundant)
- Line 78: change “of in previously identified eight genes” to “of eight previously identified genes”
- Line 176: “yeast cells with additional LEU2 gene deleted” sounds like a second LEU2 gene is deleted. Replace with “yeast cells with the LEU2 gene deleted”
- Line 178: change “capable for simultaneous 6 gene disruptions” to “capable of 6 simultaneous gene disruptions”
- Lines 221-223: this sentence is unclear and misleading. It’s already well known that knocking out activation pathways leads to accumulation of FFAs (and the further increase provided by the second group of 4 knockouts is very minor). The authors could instead state that for both FFAs and total FAs, combining the 8 knockouts gave higher levels.

REVIEWERS' COMMENTS:

Reviewer #1 (Remarks to the Author):

The authors have responded diligently to my comments and provide additional data demonstrating the performance of their system for de novo gRNA design targeting multiple genes ("real world" example). If a few minor comments are addressed (see below), I feel the manuscript could be ready for publication depending on the level of novelty required by this journal and the respective editorial policies.

1. About the new "real world" HIS example provided by the authors (their rebuttal: We thank for this constructive criticism and we agree with the reviewer that we need a real application for our Lightning GTR-CRISPR system. We therefore chose 6 HIS genes (HIS1, 2, 3, 5, 6, and 7) for an additional demonstration of our method. All gRNAs were predicted as first hits using website (<https://atum.bio/eCommerce/cas9/input>). The simultaneous disruption efficiency is 23.3% (n=6) using A2A2A2 Lightning GTR-CRISPR system. (Supplementary Fig. S6).) This information should not be hidden in the supporting information, move it to the main manuscript. I believe Nat. comm. allows for up to 10 display items, so this should be easily possible. Furthermore, this information should also be added to the abstract. The authors have currently specified in the abstract the efficiency of 87% for 8 genes "Using defined gRNAs reported earlier". Add a sentence that mentions the knockout of 6 genes using de novo designed gRNAs with 23% efficiency. Thereby readers are immediately provided with the key information. Also, it might be good to add a conclusion/summary statement on the implications of these results in the manuscript – ultimately this suggests that your system works quite well and there is rather an issue with our ability to generate efficient gRNAs against unprecedented targets. Something like "With previously functionally verified gRNAs, we could knock out up to 8 genes at 87% efficiency. Using de novo designed gRNAs, the efficiency reduced to 23% for 6 genes, whereas 8 genes could not be successfully knocked out. These results may suggest, that current gRNA design is rather a limitation than their co-expression using our lightning system. Improvements of gRNA design may hence also boost the performance of our lightning system in future applications." Update: I saw later on that the authors briefly mentioned this in the discussion section. I think it would still be good to mention the explicit results, how to come to this conclusion.

Thanks for your insights. We have included the indicated figure as Fig. 5 in the main text. We have added the results of un-optimized gRNAs (23% efficiency) in the abstract and introduction. The discussion of improving gRNA design has been added in the line 188-190.

2. About my previous comment "What happens if the 8 gRNA / knockout gene combination tested with lightning in vivo cloning? Or is this just not possible due to the loop out issues?" The authors replied: "We have tested 8 gRNAs with the lightning GTR-CRISPR system using either one plasmid with 3 different markers or two plasmids with each having 4 gRNAs. In both cases, we got either no colonies or very low efficiency (~1%) of red colonies (ADE2 disrupted). Thus, we concluded that it is not possible to perform efficient lightning GTR-CRISPR for 8 targets with the current conditions. This can be due to a loop-out issue or unsuccessful Gold-Gate reaction

(because of too many fragments). We have included a comment about this in the main text “The further extension of Lighting GTR-CRISPR system for 8 gRNAs using a similar strategy was not successful.” This explanation is sound and should be provided as part of the manuscript, I imagine this could be quite relevant to some readers. So extend the one sentence you have already written with the full explanation you have written to me (add a “data not shown” for the 1% red colonies etc.).

We have extended the statement in main text in the line 174-179.

3. About my previous comment “Why exactly 8 bp? and not something shorter but 6 or 3? Comments for “we first tested three different modes for introducing simultaneous gene disruptions (8bp frameshifting deletions around PAM sequences) of 3-5 genes in *S. cerevisiae*.” Authors’ reply: In this design, we considered that there are some reports of the potential off-target cleavage still occurring with 3-5 bp mismatches in the PAM-distal part of the gRNA sequence (Pattanayak et al Nat Biotechnol. 2013 and Hsu et al Nat Biotechnol. 2013). Thus, we deleted 8bp including the PAM sequencing to avoid potential off-targets. And we totally agree with the reviewer that the frameshifting deletion can be shorter than 8bp and we have successfully generated shorter deletions using GTR-CRISPR system in other cases. This is interesting and relevant information, please also add these references and the brief explanation to the manuscript.

We have added these information and references in the line 92-95.

4. The authors have only marked changes to the previous version with yellow highlighting. Thereby the initial text has been lost. Or it is required to tediously compare the two versions. I am not willing to do that, so my review is somewhat incomplete and the editorial teams needs to decide if they want to manually check the changes. For any future resubmissions, the authors should use the “Track changes” functionality of Word, allowing to compare new and old text.

We used “Track changes” in this resubmission.

Reviewer #2 (Remarks to the Author):

Summary:

The manuscript describes the development of new CRISPR systems for multiplexed genome engineering in the yeast *Saccharomyces cerevisiae*. The research employs gRNA-tRNA arrays that have also been shown to be efficient for this purpose in other organisms. The authors then develop a rapid cloning-free system that skips transformation and checking in *E. coli*. While the general approaches are not novel, their adaptation for the rapid knockout of multiple genes in yeast is carefully executed and optimized, and the results obtained are impressive. These are significant improvements over the currently published multiplexing approaches for *S. cerevisiae*. The methods and results will be of interest to a wide range of researchers engineering the *S. cerevisiae* genome, and to those interested in similar tools for other yeast species.

The authors have addressed the comments from the reviewers, and their response and revisions are generally satisfactory. They have included missing details, added new experimental results, and have improved the manuscript. However, a remaining concern is in the preparation – there are still significant grammatical/wording issues, including in the new text added to the manuscript:

1. In response to a reviewer comment, the authors now refer to their gRNAs as “defined gRNAs”. However, it is unclear what is meant by “defined”. The authors should explicitly state that these were optimal gRNAs previously shown to be effective, and then refer to them simply as gRNAs.

Thank you for pointing out. We have changed “defined gRNAs” to “reported gRNAs shown to be effective” when we first mentioned gRNAs.

2. New paragraph on p. 8: This needs to be carefully rewritten as there are several errors and some sentences are unclear. Two examples: “unprecedented” appears to be an incorrect word choice, and line 85: “system was still enabled simultaneous 6 gene disruptions” needs to be fixed. Also, as mentioned above, “defined gRNA” is unclear and needs to be described more completely.

Fixed.

3. Other examples of corrections needed:

- Abstract, lines 31-34: suggest changing to “by about 30-fold in 10 days using just two rounds of deletions of eight previously identified genes”
- Line 54: change “maximum” to “a maximum”
- Lines 62-63: remove either “Therefore” or “thus” (redundant)
- Line 78: change “of in previously identified eight genes” to “of eight previously identified genes”
- Line 176: “yeast cells with additional LEU2 gene deleted” sounds like a second LEU2 gene is deleted. Replace with “yeast cells with the LEU2 gene deleted”
- Line 178: change “capable for simultaneous 6 gene disruptions” to “capable of 6 simultaneous gene disruptions”
- Lines 221-223: this sentence is unclear and misleading. It’s already well known that knocking out activation pathways leads to accumulation of FFAs (and the further increase provided by the second group of 4 knockouts is very minor). The authors could instead state that for both FFAs and total FAs, combining the 8 knockouts gave higher levels.

Thanks for pointing out these mistakes. We have corrected all above points and we have corrected other mistakes in the text.